# Motion fluency effects on object preference is limited to learned context

**Jonathan Charles Flavell**⬤*°, **Bryony McKean**°

Department of Psychology, University of York, York, North Yorkshire, United Kingdom

° These authors contributed equally to this work.
* jonathan.flavell@york.ac.uk

## Abstract

Recently, Flavell et al. (2019) demonstrated that an object's motion fluency (how smoothly and predictably it moves) influences liking of the object itself. Though the authors demonstrated learning of object-motion associations, participants only preferred fluently associated objects over disfluently associated objects when ratings followed a moving presentation but not a stationary presentation. In the presented experiment, we tested the possibility that this apparent failure of associative learning / evaluative conditioning was due to stimulus choice. To do so we replicate part of the original work but change the 'naturally stationary' household object stimuli with winged insects which move in a similar way to the original motions. Though these more ecologically valid stimuli should have facilitated object to motion associations, we again found that preference effects were only apparent following moving presentations. These results confirm the potential of motion fluency for 'in the moment' preference change, and they demonstrate a critical boundary condition that should be considered when attempting to generalise fluency effects across contexts such as in advertising or behavioural interventions.

## Introduction

This subjective ease or difficulty of processing perceptual information is known as perceptual fluency [1] and there is considerable evidence demonstrating that the easier the perceptual processing of a target, the more positive the assessment of that target is. Well known fluency factors include target contours (e.g. [2,3]), symmetry (e.g. [3,4]). background contrast (e.g. [5,6]), and ambiguity resolution (e.g. [5]). Many fluency manipulations are visually interesting and engaging, and some can shift preference after a single short presentation. Fluency manipulations therefore have clear potential for use in applied settings such as advertising campaigns or behavioural interventions.

However, preference effects are typically measured in the lab during or immediately after a fluency manipulation. To be truly useful the preference effect must persist to a situation in which the stimulus is presented without an ongoing manipulation i.e. object-fluency associations must be learned and later recalled when the object is seen free of its earlier fluency manipulation. This means that manipulations would ideally not alter the target's composition (e.g. a product's outline or design) and would instead influence some aspect of the target's

---

**Data Availability Statement:** Data, stimuli, statistical models, and supplementary documents are available at osf.io/htqrm.

**Funding:** This research was supported by a Leverhulme Trust (leverhulme.ac.uk) grant awarded to Steven P Tipper and Harriet Over (grant

reference no. RPG-2016-068). The funders had no role in study design, data collection and analysis, decision to publish, or preparation of the manuscript.

**Competing interests:** The authors have declared that no competing interests exist.

presentation. This was recently achieved using motion fluency. Flavell, McKean, et al. (2019) [7] demonstrated that objects repeatedly paired with fluent motion (smooth/predictable size and direction changes) were preferred over those repeatedly paired with disfluent motion (sudden/unpredictable size and direction changes). That study also demonstrated learning of object-fluency associations but, critically for applied work, the retrieval of the associations failed in situations where participants rated stationary objects. That is, though object-fluency associations were learned and led to preference effects on moving objects, these associations were not evoked when objects were presented in a stationary (i.e. non-manipulated) context.

This finding was surprising given that associative learning / evaluative conditioning (see [8] for review) predicts that learning and retrieval would have occurred following the repeated pairing of the conditioned stimulus (CS; the object)) with the positive (fluent) or negative (disfluent) unconditioned stimulus (±US; motion). It is possible that the observed failure of retrieval from associative learning following context change may be due to the stimuli that were used–household objects and geometric shapes. Animal studies of associative learning demonstrate that not all combinations of stimulus and response can be paired effectively, and that that associations are learned better when the stimulus and response are naturally related and evolutionarily relevant. For example, the learned association in rats between a flavour (the 'natural' CS) and a subsequently received electric shock (an 'unnatural' -US) is limited to the environment in which the shock and flavour were encountered, meaning that a particular flavour is only avoided in the shock environment (i.e. association is context limited). However, when flavour is instead associated with an illness producing poison (a 'natural' -US) the rat will avoid that flavour in all environments [9] (for similar findings see also [10–15]).

This suggests that the 'naturally stationary' stimuli used by Flavell, McKean, et al. (2019) [7] may be not be well suited for association with motion fluency when retrieval is in a different context. If this is the case, then it is important to establish the boundaries of such rapidly-induced preference effects. For example, in a behavioural intervention it might be less appropriate to use motion fluency to influence liking of foods than it would be liking of animals.

To explore this, we present an experiment replicating aspects of Flavell et al.'s (2019) [7] Experiments 7 and 8 but with an important change to the stimulus set. Rather than using abstractly patterned geometric shapes, here we use winged insects whose natural airborne motion is reminiscent of the movement used in the original studies. Thus, the ecological validity of the object-motion pairings should facilitate associative learning and increase the likelihood of detecting fluency effects when those objects are later presented in a stationary context.

As in the original experiments, participants in the present study were exposed to multiple pairings of objects with either fluent motion (predictable) or disfluent (unpredictable) motion. Continuous attending was encouraged by using a task in which participants had to press a button in response to a transient change in insect colour saturation (from full colour to greyscale). Furthermore, the task used a two-alternative forced choice paradigm in an attempt to bolster associative learning by deepening encoding of each object's identity. That is, participants must actively identify the target to retrieve the correct key assignment. This means that object perception must pass preliminary stages concerned with low-level sensory information such as brightness and later stages concerned with pattern recognition and pairing the input of insect type against the learned associations of the insect's motion [16]. Unlike preliminary processing, deeper levels of processing are stored in memory, so by targeting these we expect a richer and stronger memory trace associated with the objects than would be expected using a single button response task.

In the present experiment, participants rated presented insects for liking before and after an exposure phase in which the insects were repeatedly and consistently paired with either fluent or disfluent motion. During ratings, each object was either stationary or moving as it did

during the exposure phase (fluently or disfluently). In this way we can explore whether fluency effects strengthen following repeated exposure and whether fluency effects survive a change from a moving to a stationary context when stimuli appropriate to the motion are used.

This experiment also serves as the first attempt to replicate the original motion fluency study of Flavell et al. 2019 [7]. Such extended replications are necessary to establish the boundary conditions for new effectors [17]. The original study demonstrated limitations to context generalisation (a failure to transfer to stationary presentation) which we attempt to overcome here. Whether preference effects are present in the stationary presentation condition determines whether the current paradigm is appropriate for an applied setting or whether further development is necessary.

## Materials and methods

### Apparatus

Participants sat at a table facing a 23" touch screen monitor (HannsG (Taipei, Taiwan) HT231HPB, 1920x1080 pixels) at approximately 50cm distance with a keyboard positioned between the participant and the screen. Participants responded to task stimuli using the left and right control keys (often labelled 'Ctrl') which were coloured green and blue respectively. Stimulus presentation (60Hz) and response recording were achieved using custom scripts and Psychtoolbox 3.0.11 [18–20] operating within Matlab 2015a (The MathWorks Inc., Natick, MA) on a PC (Dell (Round Rock, USA) XPS, Intel (R) Core (TM) i5-4440, 3.1 GHz CPU, 12 GB RAM, 64 bit Windows 7).

### Stimuli and motion fluency

Four winged insects featured in the experiment. Each insect had two forms: a full colour 'standard' form (Fig 1A) and a grey scale 'catch' form (Fig 1B). Images available at osf.io/htqrm.

On a fluent trial (Fig 1C top row) the insect would appear in the centre of the screen, remain stationary for 500ms, and then move for 2500ms after which it would disappear. During that motion, the insect either expanded or contracted whilst simultaneously rotating 90˚ either clockwise or anticlockwise. On these fluent trials the rate of size change and rotation was constant.

Disfluent trials were adapted from fluent trials. Here, each motion was divided into 5 equal sections and reordered from [1–2–3–4–5] to [1–4–3–2–5] (Fig 1C bottom row). This resulted in identical retinal information for equivalent fluent and disfluent trials whilst changing the ease of processing. The expansion/contraction × clockwise/anticlockwise manipulations results in 4 possible movements for each of the fluent and disfluent conditions.

In all trials, an expanding insect's wingspan at the moment of appearance was 35mm and a contracting insect's wingspan was 130mm. The final wingspan for all insects was 83mm. The final orientation was the same for all insects (head towards top of screen as shown in Fig 1A & 1B). Further trajectory information and video examples of object movements are available at osf.io/htqrm.

### Procedure and task

The experiment consisted of a first exposure rating block, a practice block, an exposure block, and a final exposure rating block. Object-fluency assignment (see next section) dictated the fluent/disfluent assignment of each insect for a given participant. This assignment also dictated whether the insects would be moving or stationary in the rating blocks. Trial order was randomised within all blocks. Participants were able to take a short break between each block.

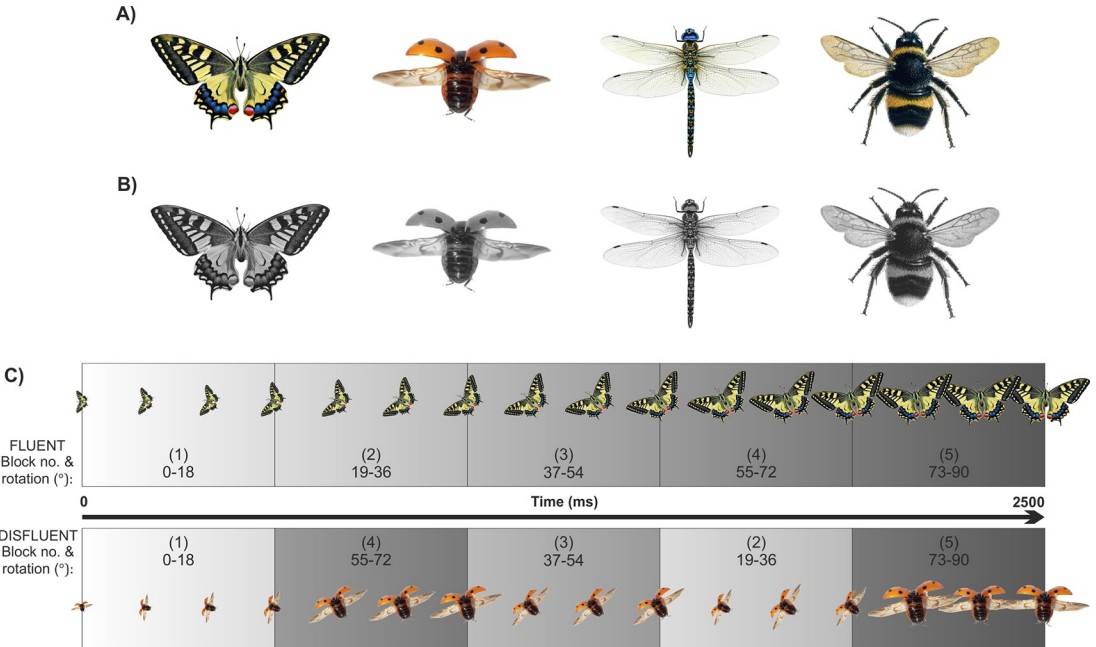

**Fig 1. Stimuli and stimuli motion.** A) 'Standard' (full colour) insects. B) 'Catch' (greyscale) insects. C) Schematic representations of object movements in the fluent (top panel) and disfluent (bottom panel) conditions. Note that the background colour in the experiments was a constant grey. In this figure the background varies to highlight the reordered sections in the disfluent condition. See osf.io/htqrm for video examples of motion.

Instructions for each block were presented on screen and verbally by the experimenter. Verbatim copies of presented instructions are available at osf.io/htqrm. The experiment took ~5 minutes to complete.

In each trial of the practice and exposure blocks an insect would appear and move either fluently or disfluently as described earlier in 'Stimuli and motion fluency'. These blocks were intended to familiarise participants with the task and allow learning of object-fluency associations. In both of these blocks participants performed a change detection task to ensure that they continually attended to the stimuli. The task was to press a keyboard button as soon as possible if the insect changed from its standard form to its catch form i.e. participants had to press a key if the object temporarily changed from colour to greyscale. Trials on which the insect changed are referred to as 'catch trials' and trials on which the insect did not change are referred to as 'standard trials'. During a catch trial, the insect changed to its catch pattern for 500ms in either section 2 or section 4 of the motion (see Fig 1B). Participants were unaware of whether a given trial was a catch trial until the insects changed colour. Readers are encouraged to watch the stimulus videos at osf.io/htqrm.

To facilitate encoding of object identity, the response key varied by insect. The bee and dragonfly were assigned to the left (green) control key and the butterfly and ladybird were assigned to the right (blue) control key. The practice block was broken down into three parts to introduce the insect-key assignments. The first part had four catch trials to pair the bee and dragonfly (2 trials each) to the green key, the second part had four catch trials to pair the butterfly and ladybird (2 trials each) to the blue key, and the final section had eight catch trials duplicating those from parts 1 and 2. Trial order was randomised in each part.

In the exposure block, each insect appeared 16 times totalling 64 trials. For each insect, eight were catch trials and eight were standard trials. Of each set of eight, half expanded, and

the half contracted. Of each set of those four trials, half rotated clockwise and half counter-clockwise.

In both the first exposure rating block and the final exposure rating block, participants separately rated each insect for liking. This rating was used to assess fluency effects. On a rating trial, an insect would first be presented and then be replaced by a 50cm long horizontal visual analogue scale presented in the centre of the screen. The scale was a continuous black line with brackets at each end and no other demarcations or ticks. Ratings on this scale were later transformed to between -100 and +100 for analysis. Participants were told to tap the scale toward the right if they liked the object or toward the left if they did not like the object, with how far left or right they tapped indicating how much they did or did not like the object. Depending on condition assignment (see later 'Object-fluency assignment'), the insect presentation was either in motion (as described in 'Stimuli and motion fluency') or stationary in the centre of the screen in its final size and orientation for 2500ms. Note that the insects never changed colour during rating trials.

## Object-fluency assignment

Participants were assigned to one of four experiment versions, which determined each insect's fluency and whether the insect would be rated following moving or stationary presentation (see Table 1). For every participant, two insects would be assigned to fluent motion and two to disfluent motion. One fluent and one disfluent insect would be rated following a moving presentation and the others rated following a stationary presentation. For example, a participant in version 1 would always see the bee and butterfly moving fluently when in motion and always see the dragonfly and ladybird moving disfluently when in motion. That same participant would see the bee and dragonfly moving during rating trials and would see the butterfly and ladybird stationary during rating trials. In combination with the first/final ratings this creates a $2 \times 2 \times 2$ design (motion/stationary × first/final × fluent/disfluent).

## Change detection task feedback

Responses on the change detection task were considered correct if they consisted of either not responding on a standard trial or pressing the correct key within the 500ms catch window on a catch trial. Correct responses were signalled to participants by a green border around the screen until the end of the trial.

Incorrect responses on the change detection task were: responding at any point on a standard trial; not responding on a catch trial; responding outside of the 500ms catch window on a catch trial; or pressing the wrong key in the 500ms catch window (e.g. pressing blue when green was assigned to that insect). Response errors were signalled to participants by all stimuli disappearing, a 100ms error tone, and a red border appearing around the screen for 1500ms. If

Table 1. Motion assignments for each experimental version.

| Version | Motion | | Static | |
| | Fluent | Disfluent | Fluent | Disfluent |
|---|---|---|---|---|
| 1 | Bee | Dragonfly | Butterfly | Ladybird |
| 2 | Ladybird | Bee | Dragonfly | Butterfly |
| 3 | Butterfly | Ladybird | Bee | Dragonfly |
| 4 | Dragonfly | Butterfly | Ladybird | Bee |

N.B. On rating trials, the moving fluent and disfluent objects always expanded and rotated clockwise in the way described earlier in *Stimuli and motion fluency*.

participants pressed the wrong key then a reminder of the insect-key assignment was shown on-screen after the error presentation. Failure-to-respond errors on catch trials were signalled to participants after the object had completed its motion and disappeared. In this case a 100ms error tone played, and a red border appeared around the screen for 1500ms.

## Data exclusion, design and analysis

To ensure that only participants who had engaged with the task were considered in analysis, those who failed to respond on >25% (>8 trials) of catch trials or responded on >25% (>8 trials) of standard trials were removed from the data set. Because participants were not pre-screened for strong preference/aversion for insects (to avoid demand characteristics), we also removed any participant whose first exposure ratings were less than -95 or greater than +95 (n = 3). Data processing was completed using custom scripts in MATLAB 2018a.

Liking ratings were analysed via generalised linear mixed-effects modelling (GLMM) in R using the package lme4. This analysis allows for variations in the baseline ratings of insects to be modelled for each participant. In other words, it accounts for participants' initial preferences. Motion and stationary ratings were modelled separately. Scale liking ratings rarely conform to normal distribution so, before analysis, the data were reflected to bring all ratings into the positive domain and a suitable distribution was then approximated. All models were subsequently specified with a gamma distribution and a log link. Each model included liking rating as the outcome variable, with 'exposure' and 'fluency' as categorical predictors (fixed effects) and participants as random intercepts. A backwards stepwise procedure was used to obtain the most parsimonious model. This procedure was implemented through manual checking and effect removal, as follows: the most complex possible model (i.e. full factorial: all main effects and all possible interactions) was run first. Non-significant effects were then removed. Effect removal took place one level at a time, as follows: if the highest-level interaction was a 2-way interaction and was not significant, it was removed and the model re-run. Each non-significant main effect was then removed in turn, and the model re-run. Non-significance was determined using Wald Chi-Square tests to compare a model to a nested version of the same model with the critical term removed.

## Participants

Protocols were approved by the University of York's Psychology Ethics Committee and were in line with the tenets of the Declaration of Helsinki. In an effort to maximise the robustness of our investigation we increased the sample size of Flavell et al. (2019) [7] by 50% to a target sample of 60 participants. Seventy-one adult participants were tested. Participants gave written informed consent prior to the experiment and were verbally debriefed at the end of the experiment. Three participants were removed from the analysis because their first exposure rating exceeded +/-95 and a further 8 participants were removed for exceeding the error threshold (mean ± SD = 12.25 ± 3.92). This left 60 participants (22 male, 36 female, 2 undisclosed gender, age mean ± SD = 20.55 ± 0.80). None of the remaining participants erred on more than 7 of 32 catch trials (mean ± SD = 2.92 ± 2.10), or on more than 2 of 32 standard trials (mean ± SD = 0.34 ± 0.65).

## Results

Liking ratings are shown in Fig 2. Modelling indicated that when ratings followed a moving presentation only motion fluency affected preference (i.e. no effect of rating block or fluency × rating block interaction). The comparison of this final model with a null model (which contained no fixed effects) was significant ($X^2$ = 9.876, $p$ = .002). Modelling also

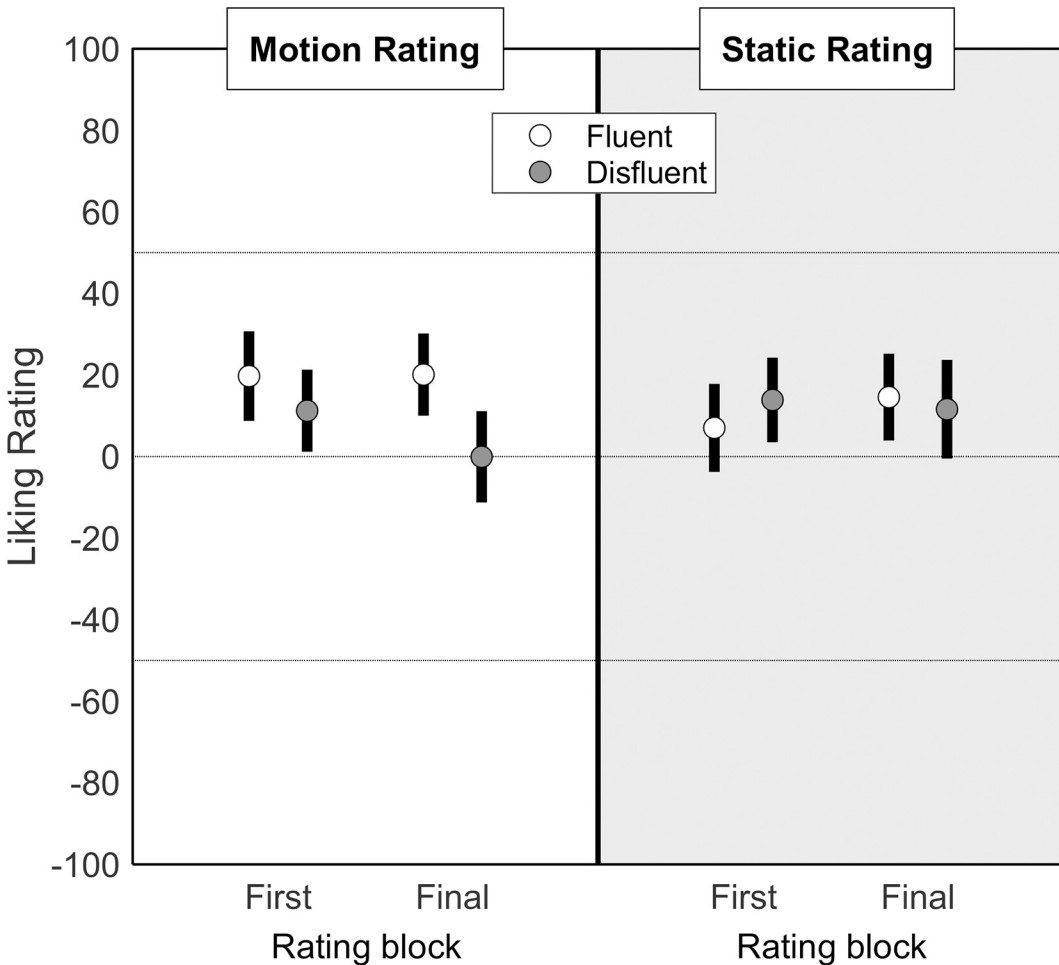

**Fig 2. Mean (±95% confidence interval) liking ratings for fluently moving (white dots) and disfluently moving (grey dots) insects.** The white panel (left) indicates ratings made following exposure to moving objects and the grey panel (right) indicates ratings made following exposure to static objects. Data are split by first / final exposure (i.e. before /after exposure to repeated object-motion pairings).

indicated that when ratings followed a stationary presentation, neither fluency, rating block nor their interaction affected preference. In other words, fluency effects on preference were evoked only when objects were rated following a moving presentation and there was no statistically observable development of that preference following multiple object-fluency pairings. Full models and comparisons available at osf.io/htqrm.

## Discussion

The presented experiment reproduces and extends Flavell et al.'s (2019) [7] study on preference effects of motion fluency. A key difference between the original study and the present one was the use of winged insects as opposed to geometric patterns, with the aim of facilitating associative learning. In an exposure phase, participants were repeatedly shown insects consistently moving in either a fluent (smooth/predictable size and direction changes) or a disfluent (sudden/unpredictable size and direction changes) manner. Participants rated either moving or stationary insects to assess the context-change-related limitations of any learned object-fluency associations. Ratings were performed before and after the exposure phase to assess the impact of those object-fluency associations.

In our introduction we suggested that the lack of motion fluency effects for stationary objects in Flavell et al. (2019) [7] was due to a limitation of associative learning. That is, the pairing of the geometric pattern to motion was not effective because such an association lacked the 'naturalness' which can facilitate learning of a pairing (e.g. [9]). We speculated that the greater 'naturalness' of insect to motion pairing in the present study would facilitate associative learning leading to preference effects for statically presented objects; however, this was not the case. Our results supported the findings of the original study: namely, that preference effects were present when stimuli were presented moving fluently/disfluently but were absent when stimuli were presented in a stationary context (i.e. in a different context to that in which the associations were learned).

A further replication of the original results comes in the lack of first/final rating effects. That is, the motion fluency manipulation (for moving targets) was sufficiently powerful to evoke a similar effect following a single presentation as it did following repeated presentations. At first glance, this may seem to indicate a failure to develop the object-fluency association. However, it is worth observing that the clear learning of object-fluency associations reported by Flavell et al. (2019) [7] using less potent motion disappeared when using the motions used in the present study. This suggests that a ceiling effect was reached after a single exposure in both sets of studies. It also suggests that we might cautiously interpret the mean differences of first/final exposure ratings (see Fig 2) as evidence of learning of object-motion associations, which under other circumstances might reach statistical significance.

Any ceiling effect might be breached if associative learning were bolstered by a greater number of object-fluency exposures. Indeed, the number of object-fluency presentations in the original study and the present study was relatively low at 16 per object. This was intentional: we wanted to keep the total experiment time short so as to be reminiscent of what might be possible in a behavioural intervention (or advertising campaign), where participant engagement is limited. However, with such concerns in mind, future investigations could explore multiple spaced exposure sessions.

The current work represents the first replication of the finding by Flavell et al. (2019) [7] that motion fluency is a potent modifier of preference but only in context in which the fluency association is learned. These results indicate an important boundary condition that should be considered when designing visuo-motor fluency applications for the real world. Overcoming this generalisation limitation appears to be a major challenge if long term or real-world preference change is to be achieved using visuo-motor fluency.

## Acknowledgments

We would like to thank Sarah Knight for statistical support, and Rebecca Cousins, Bethany Gamble, Emily Hamilton and Christopher Snape for assistance with data collection.

## Author Contributions

**Conceptualization:** Jonathan Charles Flavell, Bryony McKean.

**Data curation:** Jonathan Charles Flavell, Bryony McKean.

**Formal analysis:** Jonathan Charles Flavell.

**Investigation:** Jonathan Charles Flavell, Bryony McKean.

**Methodology:** Jonathan Charles Flavell, Bryony McKean.

**Project administration:** Jonathan Charles Flavell, Bryony McKean.

**Resources:** Bryony McKean.

**Software:** Jonathan Charles Flavell, Bryony McKean.

**Supervision:** Jonathan Charles Flavell.

**Visualization:** Jonathan Charles Flavell.

**Writing – original draft:** Jonathan Charles Flavell, Bryony McKean.

**Writing – review & editing:** Jonathan Charles Flavell, Bryony McKean.

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
