## [Decision Letter · Decision Letter 0]

21 Oct 2020

PONE-D-20-19462

Motion fluency effects on object preference is limited to learned context

PLOS ONE

Dear Dr. Flavell,

Thank you for submitting your manuscript to PLOS ONE. After careful consideration, we feel that it has merit but does not fully meet PLOS ONE’s publication criteria as it currently stands. Therefore, we invite you to submit a revised version of the manuscript that addresses the points raised during the review process.

We look forward to receiving your revised manuscript.

Kind regards,

Deborah Apthorp, Ph.D

Academic Editor

PLOS ONE

Additional Editor Comments:

Both reviewers have commented that the manuscript lacks novelty, but since this is not a criterion for publication in PLoS One, I do not require the authors to respond to that concern.

Data availability is a pre-requisite for publication in PLoS One - see https://journals.plos.org/plosone/s/data-availability. While I note that the dataset is available as a .csv on the OSF, along with a document called "models" - which is great! - there is no code for reproducing the analyses. I note that all the models refer to a dataset called "prefDataSUBSET". Is this the same as the data called "data" on the OSF? If not, how were the data subsetted? It would be great if R code could be provided, as I note the analysis was performed in R - this would enhance replicability.

In addition to this, both reviewers note some important points that need to be addressed, which are detailed below.

Journal Requirements:

Reviewers' comments:

Reviewer's Responses to Questions

**Comments to the Author**

1. Is the manuscript technically sound, and do the data support the conclusions?

Reviewer #1: Partly

Reviewer #2: Yes

2. Has the statistical analysis been performed appropriately and rigorously? 

Reviewer #1: Yes

Reviewer #2: Yes

3. Have the authors made all data underlying the findings in their manuscript fully available?

Reviewer #1: Yes

Reviewer #2: Yes

4. Is the manuscript presented in an intelligible fashion and written in standard English?

Reviewer #1: Yes

Reviewer #2: Yes

5. Review Comments to the Author

Reviewer #1: This paper reports an experiment to examine the effect of motion fluency on preference judgement, by using a (relatively) ecological-valid images of insects. The results showed that repeated presentation of fluent or disfluent motion affected later preference judgement only when the test stimuli were also moving. The authors claim that the results replicated their earlier findings.

The major problem is that the messages from the current study is limited to partial replication of their own paper. I do not mean that replication is trivial, but that its own message may not be sufficient. Also, I mean 'partial' because the results did not show the effect of preference by fluency, but it rather seems to show decrease of liking by disfluency (even though it could be referred to as the effect of fluency anyway). This point is not fully discussed, just implied by 'the lack of first/final rating effects' (L296). The effect was significant only as a form of interaction, and the disfluency effect may not be significant, but it seems to be a crucial point if the authors suggest applications of this effect (L16-17) and should therefore be discussed more thoroughly with statistical supports.

Minor comments

The abstract should represent the current study properly on its own. Now most of the abstract describe the previous results. Many readers will not continue to the main part, as they simply take it as a pure replication of their previous study, but actually there are some novel points in the methods that supplement, not just replicate, their earlier findings.

The introduction should also begin as independent of the abstract.

As the authors are aware, insects are quite aversive for quite some people, and therefore may not be appropriate for the current purpose. A ladybird or a bee of 13 cm wide could be quite disturbing even for those who do not have insect phobia. So I wonder if there might have been difference in the results for expanding and contracting stimuli. Also, the number of participants who exceeded +95% and -95% should be separately noted.

L183: It is not very clear, but if the response line was continuous and had no ticks, we do not call it a Lickert scale. Maybe better referred to as a visual analogue scale. (Please check.)

Reviewer #2: This is a study about visual perception and preference. There is a theory that perceptual fluency affects how much people like after a brief presentation. The starting point is a previous work by the same authors. In that study, it was the smooth and fluent motion of the object that affected preference. The main novelty in this study is the use of stimuli which are not abstract but rather are images of winged insects. Observers (N=60) reported levels of linking before and after seeing the motion for each stimulus. The results were consistent with the previous findings, in that fluency effects on preference were evoked

only when objects were rated following a moving presentation.

The paper is well written. The figures are nice and clear. The study has enough power to allow an interpretation also of a null finding. On the other hand, this is still a single study with fundamentally one change compared to previous studies. It is in this sense incremental and perhaps more manipulations could have been tested. Although the insects look realistic, their motion is not very ecological. That is it is far from the actual motion of a bee or butterfly. The most likely interpretation in my view is that the liking effect is directly a response to the motion, irrespective and therefore unrelated to the object. Despite these limitations, and considering how difficult it is to test additional participants during a pandemic, my overall assessment of the study is positive.

6. PLOS authors have the option to publish the peer review history of their article (what does this mean?). If published, this will include your full peer review and any attached files.

Reviewer #1: No

Reviewer #2: **Yes: **Marco Bertamini

---

## [Author Response · Author response to Decision Letter 0]

12 Nov 2020

Dear Professor Apthorp,

We would like to thank you and the reviewers for the time and effort put into the reviews. We have made a number of changes, in line with the reviewer’s comments. Reviewer comments are presented below in black and our responses in blue. Revisions to the manuscript text are made with track changes.

Yours sincerely,

Jonathan Flavell and Bryony McKean

Response to Editor comments

While I note that the dataset is available as a .csv on the OSF, along with a document called "models" - which is great! - there is no code for reproducing the analyses. I note that all the models refer to a dataset called "prefDataSUBSET". Is this the same as the data called "data" on the OSF? If not, how were the data subsetted? It would be great if R code could be provided, as I note the analysis was performed in R - this would enhance replicability.

The R code (JF_code.R) used for our analysis is now added to OSF folder (osf.io/htqrm). That code reads the file (JF_data.csv, also now on OSF) as prefData which contains the moving and static data. The subset is whether motion or static data are to be analysed in the subsequent models. 

The file JF_data.csv is the same as the existing data.csv but with different column headers and column order. I listed data.csv originally because the column headers were clearer and in a more sensible order than the source JF_data.csv. For completeness I have left both files on the OSF along with a note explaining the above. 

We have check the manuscript for discrepancies:

- Figure 1 was moved earlier by several paragraphs to appear directly after its first citation. 

- Author addresses have been corrected

- Figures have been reformatted as TIF

In Participants we know state that written informed consent was given prior to participation and that verbal debrief was given after participation. No minors were tested. We have added the word “adult” to the number of participants tested to make this clear.

Response to Reviewer #1

The major problem is that the messages from the current study is limited to partial replication of their own paper. I do not mean that replication is trivial, but that its own message may not be sufficient. Also, I mean 'partial' because the results did not show the effect of preference by fluency, but it rather seems to show decrease of liking by disfluency (even though it could be referred to as the effect of fluency anyway). This point is not fully discussed, just implied by 'the lack of first/final rating effects' (L296). The effect was significant only as a form of interaction, and the disfluency effect may not be significant, but it seems to be a crucial point if the authors suggest applications of this effect (L16-17) and should therefore be discussed more thoroughly with statistical supports.

Regarding “The effect was significant only as a form of interaction, and the disfluency effect may not be significant”. This is not the case. Though Figure 2 shows an apparently smaller fluent|disfluent object liking difference at the first rating than at the final rating, modelling indicated that was no interaction. As such we state “motion fluency manipulation (for moving targets) was sufficiently powerful to evoke a similar effect following a single presentation as it did following repeated presentations” in Discussion. 

We did cautiously suggest that there might be learning based on the finding of apparently larger mean differences in disfluent ratings following repeated motion exposure (end of paragraph 4 in Discussion). We are reluctant to add to that discussion because we present only a single motion exposure experiment and the greater difference may arise from noise. However, if a series of experiments presented similar data patterns using similar motions and stimuli then further investigation of this potential difference between fluency|disfluency would be warranted. 

The abstract should represent the current study properly on its own. Now most of the abstract describe the previous results. Many readers will not continue to the main part, as they simply take it as a pure replication of their previous study, but actually there are some novel points in the methods that supplement, not just replicate, their earlier findings.

 The introduction should also begin as independent of the abstract.

Thank you for pointing this out. We have made substantial changes to the abstract to better represent the current study and standalone from the Introduction. 

As the authors are aware, insects are quite aversive for quite some people, and therefore may not be appropriate for the current purpose. A ladybird or a bee of 13 cm wide could be quite disturbing even for those who do not have insect phobia. So I wonder if there might have been difference in the results for expanding and contracting stimuli. 

 The motion of fluent and disfluent objects on ratings trials was always expanding and clockwise. Full counterbalancing of this aspect was not possible with the current design which was already 2 x 2 x 2 (motion/stationary × first/final × fluent/disfluent). This has now been made clear in the Object fluency assignment.

Also, the number of participants who exceeded +95% and -95% should be separately noted.

This was reported in Participants but is now also noted in Data exclusion, design and analysis where the exclusion is first mentioned.

L183: It is not very clear, but if the response line was continuous and had no ticks, we do not call it a Lickert scale. Maybe better referred to as a visual analogue scale. (Please check.)

 The response line was continuous and had no ticks. This has been made clearer in the text. The word “Likert…” has been replaced with “Visual analogue…”. Both changes are in the final paragraph of Procedure and task.

Response to Reviewer #2

Although the insects look realistic, their motion is not very ecological. That is it is far from the actual motion of a bee or butterfly. The most likely interpretation in my view is that the liking effect is directly a response to the motion, irrespective and therefore unrelated to the object. Despite these limitations, and considering how difficult it is to test additional participants during a pandemic, my overall assessment of the study is positive.

The presented motions were taken from our previous work which demonstrated the efficacy of those motions for producing preference effects. The same motions with difference stimuli were an incremental attempt to overcome the failure the associative learning that we found earlier rather than to present a true representation of insect motion.

The reviewer suggests that preference is expressed for the motion irrespective of the object. This is an interesting idea but would require several dedicated experiments to properly explore. 

 We would like to thank the reviewer for their consideration of limitations of current testing circumstances.

---

## [Decision Letter · Decision Letter 1]

30 Nov 2020

PONE-D-20-19462R1

Motion fluency effects on object preference is limited to learned context

PLOS ONE

Dear Dr. Flavell,

Thank you for submitting your manuscript to PLOS ONE. After careful consideration, we feel that it has merit but does not fully meet PLOS ONE’s publication criteria as it currently stands. Therefore, we invite you to submit a revised version of the manuscript that addresses the points raised during the review process.

There are just a couple of minor points raised by Reviewer 1 - in particular, the wording of the Abstract - that need to be addressed before the manuscript can be accepted. 

We look forward to receiving your revised manuscript.

Kind regards,

Deborah Apthorp, Ph.D

Academic Editor

PLOS ONE

Additional Editor Comments (if provided):

Both reviewers are happy with your revisions, but Reviewer 1 still has a couple of minor points which should be straightforward to address.

Reviewers' comments:

Reviewer's Responses to Questions

**Comments to the Author**

1. If the authors have adequately addressed your comments raised in a previous round of review and you feel that this manuscript is now acceptable for publication, you may indicate that here to bypass the “Comments to the Author” section, enter your conflict of interest statement in the “Confidential to Editor” section, and submit your "Accept" recommendation.

Reviewer #1: (No Response)

Reviewer #2: All comments have been addressed

2. Is the manuscript technically sound, and do the data support the conclusions?

Reviewer #1: Yes

Reviewer #2: Yes

3. Has the statistical analysis been performed appropriately and rigorously? 

Reviewer #1: Yes

Reviewer #2: Yes

4. Have the authors made all data underlying the findings in their manuscript fully available?

Reviewer #1: Yes

Reviewer #2: (No Response)

5. Is the manuscript presented in an intelligible fashion and written in standard English?

Reviewer #1: Yes

Reviewer #2: (No Response)

6. Review Comments to the Author

Reviewer #1: In this revision, problems that I found in the original manuscript have been mostly corrected, and given the editor’s comment that novelty is not a criterion, now I have no objection in publication of this paper. I would just note relatively minor points.

1. Abstract: Much improved now, but there is just one concern. “We propose that this apparent failure of associative learning / evaluative conditioning was due to stimulus choice.” Quick readers could take this as the take-home message of this paper, which is obviously wrong as clearly stated in Discussion (L294-295). Isn’t there better wording? (such as “We tested the possibility that…”)

2. L205. N.B. While the N.B. says ‘as described earlier in Stimuli and motion fluency’, I cannot find such a description there. Actually this section does not tell anything about the rating trial, which the section should do rather than noted later.

Reviewer #2: I thank the authors for the changes in the revised manuscript. Some limitations remain and are acknowledged, but the comments have been addressed.

7. PLOS authors have the option to publish the peer review history of their article (what does this mean?). If published, this will include your full peer review and any attached files.

Reviewer #1: No

Reviewer #2: **Yes: **Marco Bertamini

---

## [Author Response · Author response to Decision Letter 1]

30 Nov 2020

Dear Professor Apthorp,

We would like to thank you and the reviewers for a second round of review. We have made changes, in line with the Reviewer #1’s comments. Reviewer comments are presented below in black and our responses in blue. Revisions to the manuscript text are made with track changes from the original document.

Yours sincerely,

Jonathan Flavell and Bryony McKean

Response to Editor comments

[no Editor comments]

Response to Reviewer #1

Abstract: Much improved now, but there is just one concern. “We propose that this apparent failure of associative learning / evaluative conditioning was due to stimulus choice.” Quick readers could take this as the take-home message of this paper, which is obviously wrong as clearly stated in Discussion (L294-295). Isn’t there better wording? (such as “We tested the possibility that…”)

Thankyou for pointing this out. We agree that this may have led some readers to incorrect assumptions about the paper. We have made changes to third and fourth sentences of the abstract in line with your suggestion. 

L205. N.B. While the N.B. says ‘as described earlier in Stimuli and motion fluency’, I cannot find such a description there. Actually this section does not tell anything about the rating trial, which the section should do rather than noted later.

Re. ‘as described in Stimuli and motion fluency’. You are correct that we do not describe the rating trial motions specifically. We were referring to the description of object motion but this was not clear so we have changed the N.B. to ‘On rating trials, the moving fluent and disfluent objects always expanded and rotated clockwise in the way described earlier in Stimuli and motion fluency.’ 

Re. ‘…does not tell anything about the rating trial…’. Stimuli and motion fluency is intended to describe the fluent and disfluent motions as clearly as possible. Object-fluency assignment is intended to describe the counterbalancing and presentation format (moving or static) of objects in task and rating trials. We described the static rating trial in the final paragraph Procedure and task because this seemed the most appropriate place for that information. We are reluctant to move that description because it would muddy the other dedicated sections. 

Response to Reviewer #2

[no Review #2 comments]

---

## [Editor Report · Decision Letter 2]

3 Dec 2020

Motion fluency effects on object preference is limited to learned context

PONE-D-20-19462R2

Dear Dr. Flavell,

We’re pleased to inform you that your manuscript has been judged scientifically suitable for publication and will be formally accepted for publication once it meets all outstanding technical requirements.

Kind regards,

Deborah Apthorp, Ph.D

Academic Editor

PLOS ONE
---

## [Editor Report · Acceptance letter]

7 Dec 2020

PONE-D-20-19462R2 

Motion fluency effects on object preferenceis limited to learned context 

Dear Dr. Flavell:

I'm pleased to inform you that your manuscript has been deemed suitable for publication in PLOS ONE. Congratulations! Your manuscript is now with our production department. 

Kind regards, 

on behalf of

Dr. Deborah Apthorp 

Academic Editor

PLOS ONE